# Context-Dependent Distinct Roles of SOX9 in Combined Hepatocellular Carcinoma–Cholangiocarcinoma

**DOI:** 10.3390/cells13171451

**Published:** 2024-08-29

**Authors:** Yoojeong Park, Shikai Hu, Minwook Kim, Michael Oertel, Aatur Singhi, Satdarshan P. Monga, Silvia Liu, Sungjin Ko

**Affiliations:** 1Division of Experimental Pathology, Department of Pathology, School of Medicine, University of Pittsburgh, Pittsburgh, PA 15261, USA; yop17@pitt.edu (Y.P.); shikai@pitt.edu (S.H.); minwook@pitt.edu (M.K.); mio19@pitt.edu (M.O.); smonga@pitt.edu (S.P.M.); shl96@pitt.edu (S.L.); 2School of Medicine, Tsinghua University, Beijing 100084, China; 3Pittsburgh Liver Research Center, University of Pittsburgh School of Medicine, Pittsburgh, PA 15261, USA; singhiad@upmc.edu; 4Division of Anatomic Pathology, Department of Pathology, School of Medicine, University of Pittsburgh, Pittsburgh, PA 15261, USA; 5Division of Gastroenterology, Department of Medicine, School of Medicine, University of Pittsburgh, Pittsburgh, PA 15261, USA

**Keywords:** liver cancer, bile duct, YAP1, trans-differentiation, mixed hepatocellular carcinoma–cholangiocarcinoma, plasticity, NRAS

## Abstract

Combined hepatocellular carcinoma–cholangiocarcinoma (cHCC-CCA) is a challenging primary liver cancer subtype with limited treatment options and a devastating prognosis. Recent studies have underscored the context-dependent roles of SOX9 in liver cancer formation in a preventive manner. Here, we revealed that liver-specific developmental *Sox9* elimination using *Alb-Cre;Sox9^(flox/flox)^* (LKO) and *CRISPR/Cas9*-based tumor-specific acute *Sox9* elimination (CKO) in SB-HDTVI-based *Akt-YAP1* (AY) and *Akt-NRAS* (AN) cHCC-CCA models showed contrasting responses. LKO abrogates the AY CCA region while stimulating poorly differentiated HCC proliferation, whereas CKO prevents AY and AN cHCC-CCA development irrespective of tumor cell fate. Additionally, AN, but not AY, tumor formation partially depends on the *Sox9-Dnmt1* cascade. SOX9 is dispensable for AY-mediated, HC-derived, LPC-like immature CCA formation but is required for their maintenance and transformation into mature CCA. Therapeutic *Sox9* elimination using the *OPN-CreERT2* strain combined with inducible *Sox9* iKO specifically reduces AY but not AN cHCC-CCA tumors. This necessitates the careful consideration of genetic liver cancer studies using developmental Cre and somatic mutants, particularly for genes involved in liver development. Our findings suggest that SOX9 elimination may hold promise as a therapeutic approach for a subset of cHCC-CCA and highlight the need for further investigation to translate these preclinical insights into personalized clinical applications.

## 1. Introduction

Combined hepatocellular carcinoma–cholangiocarcinoma (cHCC-CCA) represents a rare and intriguing entity in the spectrum of primary liver cancers, characterized by the coexistence of hepatocellular and cholangiocellular differentiation within the same tumor [1,2,3]. This dual histological phenotype poses significant diagnostic, prognostic, and therapeutic challenges, reflecting the complex interplay between hepatocytic and biliary lineages in liver tumorigenesis [4,5]. Especially noteworthy is the distinct response of this tumor type to broad-spectrum therapeutic and immune therapies, making the therapeutic strategy largely reliant on basic histological observations, such as measuring the ratio of respective tumor types [6]. Despite its clinical and pathological significance, the underlying molecular mechanisms driving the development and progression of cHCC-CCA remain poorly understood.

Traditionally, HCC and CCA are thought to originate from hepatocytes (HCs) and cholangiocytes (biliary epithelial cells; BECs), respectively, as evidenced by typical cellular morphology, unique structures, and the expression of cell type-specific markers. However, the frequent detection of human CCA and cHCC-CCA in the pericentral area of the liver lobule, a region anatomically different from native biliary structures, as well as the documented occurrence of HC-to-BEC differentiation in various chronic cholestasis models, has led to speculation that hepatocytes may also be the origin of a subset of human CCA and cHCC-CCA [4,7,8].

Indeed, numerous studies have provided evidence supporting this theory by inducing HC-derived cHCC-CCA using the HC-specific co-expression of proto-oncogenes and biliary lineage commitment genes, such as myristoylated *Akt* (*Akt*) and constitutive-active YAP1 or NRAS [9,10,11,12]. The overexpression of these oncogenes successfully produces separate regions of HNF4α^+^;panCK^+^;CK19^−^ poorly differentiated HCC and HNF4α^−^;CK19^+^ CCA, resembling the clinical cHCC-CCA tumor pathology [9,10,13].

Recently, Liu et al. reported that the biliary-specific transcription factor SOX9 determines the fate of YAP1 alone-dependent murine liver cancer; the chronic deletion of SOX9 in HC suppresses YAP1-mediated CCA-like tumor formation while promoting aggressive HCC tumor development, suggesting SOX9 as a major commitment of YAP1-mediated liver cancer lineage [14]. However, whether SOX9 is required to maintain the biliary fate of CCA within fully developed advanced cHCC-CCA remains elusive. Additionally, it is unclear whether SOX9 removal directly converts CCA lesions into HCC fate or simply eliminates CCA tumors during tumor formation, thereby allowing HCC to remain in YAP1-independent cHCC-CCA settings. Furthermore, given the discrepancy between chronic and acute gene deletion in diseased liver [15], the effects of chronic versus acute SOX9 elimination on the development of HC-derived cHCC-CCA remain to be thoroughly investigated. 

Herein, we reveal that unlike chronic developmental *Sox9* deletion, the acute and therapeutic elimination of *Sox9* reduces overall cHCC-CCA tumor burden in *Akt-YAP1* but not the *Akt-NRAS* model. Additionally, we found that transcriptional repressor DNA methyltransferase1 (DNMT1) is partially involved in the SOX9-dependent maintenance of *Akt-NRAS* cHCC-CCA. These findings underscore the context- and stage-dependent distinct roles of SOX9 in liver cancer, with potential biological and therapeutic implications.

## 2. Materials and Methods

### 2.1. Mouse Husbandry and Breeding

All animal care and experiments were performed in accordance with the Institutional Animal Care and Use Committee (IACUC) at the University of Pittsburgh. *Sox9^(flox/flox)^* and *Albumin-Cre* mice were purchased from Jackson Laboratories for breeding. All transgenic and KO mouse lines were maintained on the immunocompetent C57BL/6 genetic background. All animals ranged from 6 to 12 weeks in age for analysis and were from either gender.

### 2.2. Patient Data

Study approval for all human tissue samples was obtained from the University of Pittsburgh (IRB# STUDY19070068). All samples were provided by the Pittsburgh Liver Research Center’s (PLRC’s) Clinical Biospecimen Repository and Processing Core (CBPRC), supported by P30DK120531. 

TMAs were constructed from archival formalin-fixed paraffin-embedded tissue blocks from 108 cholangiocarcinoma patients seen at the University of Pittsburgh Medical Center and were also obtained from PLRC’s CBPRC. All tumor hematoxylin and eosin (H&E)-stained slides were reviewed, and representative areas were carefully selected for tissue microarray construction. Two random 1.0 mm sized cores were punched from each patient’s tumor and harvested into recipient blocks. The demographics and additional information about these cases are included in Appendix A. The TMAs were stained manually using an antibody against SOX9 (EMD Millipore, Burlington, MA, USA, 01803), YAP (Cell Signaling, Danvers, MA, USA, 01923), as described in the IHC sections. Whole-slide image capturing of the tissue microarray was acquired using the Aperio XT slide scanner (Leica Biosystems, Deer Park, IL, USA, 60010). The staining was evaluated and scored by an anatomic pathologist (A.S.). Staining for SOX9 and YAP was scored either as 0 (negative), 1+ (mostly cytoplasmic staining or very weak staining in CC tumor cells), or 2+ (strong positive nuclear staining in CC tumor cells). For SOX9 and YAP1, the scores for different tissue sections from each patient were averaged to obtain a single score per patient (Appendix A). Mean scores greater than or equal to 1.5 were considered “HIGH”, and mean scores less than 1.5 were considered “LOW/NEGATIVE”.

## 3. Results

### 3.1. Forced Expression of Myristoylated Akt and YAP1 in HC Yields cHCC-CCA, While the Chronic Elimination of Sox9 Induces Molecular Phenotype Switch to an Aggressive HCC

Previously, it has been reported that the sleeping beauty transposon/transposase-hydrodynamic tail vein injection (SB-HDTVI) delivery of *myristoylated Akt (Akt)* and *YAP1 S127A (YAP1)* induced HC-derived cHCC-CCA in a Notch-dependent manner [10,13]. Tumor-specific *Notch2* deletion switched the tumor type in the *Akt-YAP1* model from cHCC-CCA to a benign hepatocellular adenoma-like tumor at the expense of CCA [10]. Moreover, HC-specific *Sox9* deletion prevents YAP1 alone-mediated HC transformation into BEC-like HCC, while it provokes pure but aggressive HCC, indicating the critical role of *Sox9* in YAP1-driven HC plasticity into biliary lineage [14]. Given that *Sox9* is a well-established direct target of NOTCH2 [16,17], we aimed to investigate the effects of cell-autonomous Sox9 deletion on lineage commitment in Akt-YAP1-driven cHCC-CCA tumorigenesis. *Akt* and *YAP1* plasmids were co-delivered by SB-HDTVI into *Alb-Cre;Sox9^flox/flox^* (*Sox9* LKO) or *Sox9^flox/flox^* (LWT) in which *Sox9* was deleted HC and BEC initiated at around day E15 [18], resulting in chronic developmental deletion prior to tumor formation (Figure 1A). Notably, *Akt-YAP1* in *Sox9* LKO led to significantly decreased survival and a much greater and more lethal tumor burden, as seen by significantly greater LW/BW and macroscopically, as compared to the *Akt-YAP1* in LWT mice (Figure 1B–D). Microscopically, in a representative tiled image of a lobe, the *Sox9* LWT livers in the *Akt-YAP1* model showed more intensely CK19-positive CCA nodules scattered throughout (Figure 1E). At higher magnification, the tumors contained separated clusters and showed CCA components which were positive for SOX9, YAP1, and panCK and an HCC component with nuclear HNF4α (Figure 2B). However, the entire *Akt-YAP1 Sox9* LKO livers were full of circumscribed tumor foci which were negative or very weak for CK19 (Figure 1E). At higher magnification, the tumors revealed poorly differentiated HCCs that lacked SOX9, expressed HNF4α, and had small scattered subsets of YAP1 and panCK-positive cells with liver progenitor cell (LPC) characteristics (Figure 2B). The quantification of CK19 staining in the tiled image (Figure 1E) verified a significant decrease in the CK19-positive area in *Sox9* LKO as compared to the LWT (Figure 1F). Further, tumors in the *Sox9* LKO *Akt-YAP1* model revealed the significantly lower expression of biliary markers, such as *Krt19* and *Epcam*, along with the expected loss of *Sox9*. They also had increased expression levels of HC-specific markers, such as *Hnf4α* and *Tryptophan 2,3-Dioxygenase* (*Tdo2*), compared to LWT, suggesting a shift from CCA to HCC (Figure 2C,D). Together, these data support the role of *Sox9* in commitment to the CCA phenotype in *Akt-YAP1*-driven cHCC-CCA tumorigenesis. Next, we examined cHCC-CCA (n = 32) in the available patient TMA for SOX9 and YAP1 localization. While the majority of the mixed tumors were concurrently positive for both markers (28/32 or 87.5%), a small subset was positive for YAP1 but negative or low for SOX9 (4/32 or 12.5%) (Figure 2E). While the numbers are too low to determine the impact on overall tumor behavior or prognosis in these cases, this observation does lend clinical credibility to our preclinical observation.

### 3.2. Transcriptomic Analysis of HCC in Akt-YAP1 Model in Sox9-LKO Reveals Significant Similarity to a Subset of Human HCCs

To directly address the clinical relevance of the HCCs observed in the *Akt-YAP1* model in the absence of *Sox9*, we performed RNA-Seq analysis (GEO accession ID: GSE200472). When comparing the LWT and *Akt-YAP1 Sox9*-LKO livers, 525 genes were upregulated and 199 genes were downregulated by FDR = 5% and absolute log2 fold change of 1 (Figure 3A). To interpret the biological functions of these 724 DEGs, pathway enrichment analysis was performed by Ingenuity Pathway Analysis. Fifty-three pathways were significantly enriched in the Akt-YAP1 Sox9 LKO livers. To determine if the mouse model mimics a subtype of HCCs in patients, the LIHC cohort of the TCGA database was analyzed using a similar pipeline [19]. When comparing 50 normal or normal adjacent control livers and 374 HCCs, the DEGs were enriched in 59 pathways. Ten pathways were commonly altered in mouse and human tumors (Figure 3B). To directly compare mouse and human studies, the 724 DEGs from the *Akt-YAP1 Sox9* LKO livers were converted to human homologous genes by the Mouse Genome Database [20]. Human expression data had 546 of the 724 DEGs, and using abs(log2FC) > 1 and FDR < 0.05, 118 of the 546 DEGs were referred to as the *Akt-YAP1 Sox9* LKO (null) or AY signature and applied to the TCGA HCC (Figure 3C). These genes could clearly separate the human normal (orange bar) and HCC (light green bar) groups very well. Lastly, NTP analysis was performed using the *Akt-YAP1 Sox9* LKO signature [21]. In the TCGA cohort, the AY signature captured 12% of HCC. This subgroup of patients is enriched in the S1 class [22] (28/46 vs. 82/328 in rest of patients, *p* = 0.0015) and has a CCA-like signature [23], (22/46 vs. 102/328, *p* = 0.03) (Figure 3D). Altogether, the HCC in the *Akt-YAP1 Sox9* LKO model recapitulates a subset of human HCC.

### 3.3. SOX9 Is Dispensable for Akt-YAP1-Mediated LPC-like Immature CCA Nodule Formation but Required for Their Transformation into Mature CCA

To explore the stage-specific roles of SOX9 in *Akt-YAP1*-driven HC-to-CCA lineage reprogramming, we carefully examined the liver histology in both LWT and *Akt-YAP1 Sox9* LKO mice. We performed serial section immunostaining to detect HC and BEC markers, including HA-tag (AKT), YAP1, SOX9, HNF4α, panCK, and CK19, at 2 weeks and 5 weeks post-HDTVI, which correspond to the stages before the clonal expansion of transduced cells and after their transformation into mature CCA, respectively (Figure 4B,C) [13]. Importantly, LPC-like immature CCA nodules with *Akt-YAP1* transduction (HA-tag^+^; nuclear YAP1^+^) were observed in both LWT and *Sox9*-LKO livers. These nodules retained an intermediate LPC morphology and co-expressed the HC marker HNF4α and the BEC/LPC marker panCK, while the presence or absence of SOX9 differed between the respective livers (Figure 4B, black dashed lines). Notably, none of these nodules in *Sox9*-LKO livers were positive for the mature BEC marker CK19, whereas a subset of panCK^+^ CCA nodules in LWT livers were CK19^+^, indicating further reprogramming into the biliary lineage with SOX9 (Figure 4C). At 5 weeks post-HDTVI, these nodules further transformed into CK19^+^ CCA nodules only in LWT livers. In contrast, panCK^+^;CK19^+^ CCA nodules were not detected in any *Sox9*-LKO livers at the same stage (Figure 4C). These data suggest that while SOX9 is not necessary for YAP1-driven HC dedifferentiation into LPCs or the initiation of biliary reprogramming, it is crucial for the maintenance and maturation of these CCA-like tumor nodules into a mature CCA lineage. This observation highlights the potential stage-dependent and distinct roles of SOX9 in fully developed *Akt-YAP1*-CCA at advanced stages, particularly from a therapeutic perspective.

### 3.4. Distinct Roles SOX9 in Akt-YAP1-Driven HCC Tumor in Regulating Proliferation

Since *Sox9* deletion decreases survival in *Akt-YAP1* mice with a larger tumor burden, we next sought to evaluate tumor cell death and proliferation through histologic observation. To investigate, we analyzed LWT and *Akt-YAP1 Sox9* LKO liver tissue at 5 weeks post-HDTVI using immunofluorescence for Ki-67 to assess proliferation and IHC for TUNEL to measure cell viability (Figure 5A). Interestingly, HCC tumors lacking SOX9 showed a significant increase in the number of Ki-67^+^;HA-tag^+^ tumor cells compared to WT (Figure 5B,C), whereas there was no significant difference in cell death between the absence of SOX9 and WT (Figure 5D,E). These findings suggest that *Sox9* deletion exacerbates *Akt-YAP1*-mediated CCA-like tumors and promotes the proliferation of HCC, leading to a significantly larger tumor burden and reduced survival. This implies a distinct role for SOX9 in liver cancer lineage.

### 3.5. Tumor-Specific Acute Sox9 Loss Represses YAP1 or NRAS-Dependent cHCC-CCA Development

Recently, there have been several publications demonstrating phenotypic differences between chronic developmental gene deletion in the liver using the *Albumin (Alb)-Cre* strain and acute gene deletion mediated by the *AAV8-Tbg-Cre/CRISPR-Cas9* system [24,25], suggesting an adaptation of HC and BEC against developmental gene deletion. Notably, *Sox9* deletion in HC and BEC from E15 days using *Alb-Cre;Sox9^(f/f)^* animals results in delayed bile duct formation [26,27], indicating a compensatory mechanism. This suggests a potential disparity between acute *Sox9* elimination and the developmental adaptation observed in *Alb-Cre;Sox9^(f/f)^* animals [26,27]. These observations prompted us to investigate the effect of *Sox9* deletion concurrently with oncogenic events in the malignant transformation of HCs. To address this, we utilized an SB-based *CRISPR/Cas9*-mediated inducible *Sox9* deletion vector system applicable via HDTVI delivery (Figure 6B). We employed two validated HC-derived cHCC-CCA models driven by two different combinations of oncogenes. We delivered *SB-STOP^(flf)−^Cas9-sg-Sox9* and *Cre* expression plasmids to specifically eliminate *Sox9* in tumors along with *Akt-YAP1* or *Akt-NRAS* to induce cHCC-CCA in the presence (*Sox9* CWT) or absence of *Sox9* (*Sox9* CKO).

Additionally, we previously reported the indispensable roles of DNMT1 in repressing the transcription of HC-specific transcription factors, which is crucial for the biliary fate commitment during Notch- or YAP1-mediated CCA cancer formation. We also showed that DNMT1 inhibition eliminates the *Akt-YAP1*-CCA region while leaving the HCC region intact, indicating that DNMT1 is required for AKT-YAP-mediated HC-to-CCA reprogramming but is dispensable for HC transformation into HCC. Since the effect of Sox9 deletion in the AY model is similar to that of DNMT1 inhibition, we investigated whether there is any crosstalk between SOX9 and DNMT1 in the *Akt-YAP1* and *Akt-NRAS* models. To explore the involvement of DNMT1 in the *Sox9*-mediated reduction in *AY/AN*-cHCC-CCA, we delivered a plasmid expressing full-length *Dnmt1* into Sox9 CKO mice to investigate its association with SOX9 during liver cancer development. We carefully evaluated tumor formation and examined tumor characteristics at 5 weeks post-HDTVI (Figure 6A). All CWT mice developed a significant burden of liver cancer, with liver LW/BW reaching 30 for *Akt-NRAS* (Figure 6C,E) and 8 for *Akt-YAP1* (Figure 6D,F) animals, respectively. Remarkably, *Cas9*-mediated acute *Sox9* disruption significantly suppressed both *Akt-YAP1* and *Akt-NRAS*-driven cHCC-CCA development regardless of *Dnmt1* delivery, as all *Sox9* CKO mice remained healthy at 5 weeks when they were sacrificed for comparison to CWT mice. Grossly, *Akt-YAP1 Sox9* CKO or *Akt-NRAS Sox9* CKO mice showed only rare tumors and significantly lower LW/BW ratios compared to the widespread gross disease in CWT (Figure 6C–F). Microscopic observation also depicts the significant decrease in HA-tag^+^ *Akt-NRAS* and *Akt-YAP1* cHCC/CCA tumors in *Sox9* CKO livers, supporting gross and LW/BW observations (Figure 6G,I). However, *Dnmt1* re-expression in *Sox9* CKO livers slightly but significantly restored tumor formation driven by *Akt-NRAS* (Figure 6H), but not *Akt-YAP1* (Figure 6J), implying the partial involvement of *Dnmt1* in *Akt-NRAS* tumor development under the SOX9. Importantly, SOX9 IHC confirmed effective elimination both in *Sox9* CKO and *Sox9* CKO-*Dnmt1* livers, indicating successful *Sox9* deletion using our plasmid system. Together, in contrast to *Alb-Cre* strain-driven developmental *Sox9* removal, tumor-specific acute *Sox9* elimination robustly prevents *Akt-YAP1-* or *Akt-NRAS*-mediated cHCC/CCA formation, irrespective of the lineage of liver cancer, while *Dnmt1* is partially responsible for *Sox9* contribution in *Akt-NRAS* liver cancer development. These data may imply the existence of the adaptive compensation of HCs against liver-specific developmental deletion of *Sox9*, which induces a distinct response against *Sox9* elimination, necessitating the examination of the therapeutic potential of *Sox9* targeting at an advanced stage.

### 3.6. Therapeutic Deletion of Sox9 Significantly Reduces Advanced Akt-YAP1 Liver Cancer

To assess the therapeutic effect of *Sox9* deletion in advanced *Akt-YAP1* or *Akt-NRAS* cHCC/CCA, we employed *Osteopontin (OPN)-CreERT2* strains, a well-validated BEC-specific Tamoxifen (TM)-inducible *Cre* expression system [28]. As previously confirmed by other groups, the administration of triple intraperitoneal (i.p.) injections of 100 mg/kg of TM effectively induces the *Cre*-mediated deletion of the *Stop* cassette, along with the floxed forms of *Tdtomato reporter*, as evidenced by the absence of RFP in the corresponding CCA and cholangiocyte (manuscript submitted elsewhere). Given the widespread expression of SOX9 in fully developed *Akt-YAP1* or *Akt-NRAS* CCA regions as well as poorly differentiated HCC regions (Figure 2B), we delivered *SB-STOP^(flf)−^Cas9-sg-Sox9* along with the *Akt-YAP1* or *Akt-NRAS* plasmid into *OPN-CreERT2* mice to induce liver cancer. We then injected 100 mg/kg of TM (i.p.) 3–4 weeks after HDTVI, for a total of three injections, to achieve the tumor-specific elimination of *Sox9* when liver cancer was fully developed at an advanced stage (*Sox9* iKO). This was followed by assessments at 6–8 weeks post-HDTVI (Figure 7A). As a control, we injected TM into the WT littermates or corn oil into the *OPN-CreERT2* strains (*Sox9* iWT) that were also injected with the same dose of *Akt-YAP1* or *Akt-NRAS* along with the *SB-STOP^(flf)−^Cas9-sg-Sox9* plasmid and sacrificed at the same stage as the experimental group with *Sox9* iKO mice. Consistently, the *Akt-YAP1* or *Akt-NRAS* iWT mice developed significant liver cancer, with LW/BW reaching 10 and 12, respectively (Figure 7B–E). Interestingly, therapeutic *Sox9* iKO significantly decreased *Akt-YAP1* tumor burden at 6–8 weeks post-HDTVI (Figure 7B,C), bringing the LW/BW ratios down to 4–5, which is comparable to mice without tumors, suggesting a strong therapeutic effect in *Akt-YAP1*-driven liver cancer. However, there was no significant tumor reduction in *Akt-NRAS* Sox9 iKO, with variable gross tumor presence and LW/BW ratios comparable to those of iWT mice (Figure 7D,E). Notably, a subset of CCA-like tumors in iWT liver displayed SOX9^−^ nodules, suggesting evident leakiness of the floxed Stop cassette in our system. Despite this, the significant and robust reduction in HA-tag^+^ liver cancer specifically in *Akt-YAP1 Sox9* iKO livers indicates the context- and oncogenic driver-dependent therapeutic potential of *Sox9* elimination in advanced *Akt*-driven cHCC-CCA tumors (Figure 7F).

## 4. Discussion

Primary liver cancer, including HCC and CCA, arises from malignancies of hepatic parenchymal epithelial cells, HC and BEC, derived from the same parental cell, the hepatoblast, during development [29]. Interestingly, the liver exhibits a remarkable capacity for regeneration, characterized by the cellular plasticity of these two adult cell populations, involving diverse and complex molecular signaling pathways [30,31]. Particularly, crucial lineage-specific transcriptional regulatory components such as SOX9, HNF4α, HIPPO-YAP1, and HNF1α/β play important roles in forced cellular reprogramming both in vitro and in vivo [29,32,33,34]. Indeed, several groups have demonstrated the translation of this plasticity into cancer settings, especially in mixed HCC/iCCA and/or cHCC-CCA tumors [13,14,35,36,37,38]. Mixed HCC-iCCA and/or cHCC/CCA are also evident in clinical practice, and the cellular and molecular basis remains unknown [39]. The *Akt-YAP1* but not the *Akt-NICD* model displayed such combined tumors, suggesting distinct roles of YAP1 and Notch signaling in cooperating with AKT activation in hepatobiliary tumorigenesis [13,40,41]. The activation of YAP1 appears to drive a more hepatoblast/LPC-like cell fate which then evolves into either CCA or HCC [37,42,43]. Importantly, YAP1 activation in conjunction with active-β-catenin yielded hepatoblastoma in the SB-HDTVI model [44]. Additionally, SOX9 was critical in directing cholangiocyte and eventually CCA cell fate in the *Akt-YAP1* model since chronic *Sox9* LKO drove the *Akt-YAP1*-reprogramed cell towards HC at the expense of biliary fate. Indeed, SOX9 has been shown to be essential for proper bile duct differentiation during hepatic development [15,26]. What was also unexpected was that the elimination of *Sox9* in the *Akt-YAP1* models not only prevented the development of the CCA component of the cHCC-CCA tumors, but it also led to a more aggressive HCC with higher proliferative index. This suggests that in the context of YAP1 activation, SOX9 may be restricting HC or HCC cell proliferation. This is a novel observation, and while SOX9 has been shown to be both upregulated [45] or downregulated [46] by YAP1 signaling and appears context-dependent, how SOX9 restricts YAP1 signaling and suppresses cell proliferation in transformed hepatocytes remains unknown. It should be noted that *SOX9* upregulation and downregulation have both been observed in various tumors, and thus, the overall biological outcome may be tissue-dependent [47,48]. In our study, SOX9 has dual roles of not just regulating the biliary differentiation of HC to yield CCA, but also restricting YAP1-dependent HC proliferation, such that in its absence, the *Akt-YAP1*-driven HCC is more proliferative and aggressive.

A recent study led by Dr. Yang’s group demonstrated two dominant roles for SOX9 in *YAP1* alone-driven liver cancer: indispensable roles in lineage determinants for HC-to-iCCA formation and responsibility for the severity of *YAP1*-HCC using *AAV8-Cre;Sox9(f/f)*-mediated *Sox9* ablation [14]. However, we previously reported that SOX9 is dispensable in NOTCH-driven HC-to-BEC/iCCA reprogramming, whereas it is involved in tumor cell viability and proliferation in *Akt-NICD* HC-derived iCCA models [13]. This indicates context-dependent roles for SOX9 in fate control and cell viability within the mammalian liver cancer. These observations also suggest that anticipating the roles of biliary factors in clinical liver cancer settings is extremely difficult and complex due to the complicated and heterogeneous molecular signature of human liver cancers. In the current study, we mainly used the SB-HDTVI-based *Akt-YAP1* and *Akt-NRAS* HC-derived cHCC-CCA models to investigate the roles of *Sox9* in the liver cancer setting, including cHCC-CCA. Importantly, some of our observations are similar to those of *YAP1* alone-driven liver cancer studies, while a large part of our investigation demonstrates distinct responses, with multiple caveats. Consistent with observations in *YAP1* alone-mediated liver cancer settings, chronic developmental deletion by *Sox9* LKO prior to the HDTVI delivery of *Akt-YAP1* induced a switch of tumor lineage from cHCC-CCA to aggressive and poorly differentiated HCC with LPC characteristics, genetically representing a subset of clinical HCC cases. However, in contrast to *Sox9* LKO or YAP1 alone-mediated liver cancer studies, acute *Sox9* disruption using the *CRISPR/Cas9* system robustly repressed *Akt-YAP1* cHCC-CCA formation, irrespective of tumor fate.

These discrepancies raise several caveats that need to be carefully considered in murine liver cancer models. In particular, the difference between *Sox9* LKO and *Sox9* CKO may underscore the importance of hepatic adaptation against the deletion of the genes involved in hepatic competence and specification during development, *Sox9* in the current case, which may be responsible for the dependency cHCC-CCA formations. From this perspective, a comprehensive investigation of the adaptive roles for validated compensation genes will be pertinent studies to elucidate the mechanism behind these differences. Furthermore, the distinct tumor microenvironment between the HDTVI delivery of *YAP1* and non-invasive TET-ON *YAP1* expression in HC used for the *YAP1*-alone liver cancer model, along with the collaboration of constitutive active AKT, needs to be carefully compared to conclude these different observations.

Previously, we reported that the NOTCH-YAP1-DNMT1 axis plays indispensable and permissive roles in HC reprogramming into CCA, whereas DNMT1 is dispensable for the maintenance of CCA [13]. Given that *Sox9* LKO specifically abrogates the *Akt-YAP1* CCA region, similar to pharmacologic DNMT1 inhibition, we posited an association between SOX9 and DNMT1 in CCA fate commitment in our cHCC-CCA models. Interestingly, SOX9-DNMT1 is partially involved only in *Akt-NRAS* tumor formation but not in *Akt-YAP1* tumor development. This may suggest that DNMT1 and SOX9 are independently regulated under YAP1-driven HC-originated CCA tumor region, while the NRAS-SOX9-DNMT1 signaling cascade may be active in *Akt-NRAS* CCA tumor cells. However, further functional validation, including examining the association of YAP1 or SOX9 with pharmacologic/genetic DNMT1 regulation on *Akt-NRAS* CCA, will be an interesting focus for future studies.

Importantly, we observed the successful formation of SOX9^−^ LPC-like immature CCA nodules in *Akt-YAP1 Sox9* LKO livers at an early stage at 2 weeks post-HDTVI, expressing equivalent markers to SOX9^+^ CCA-like nodules in Sox9 WT livers, which is similar with the case of *Akt-NICD Sox9* KO livers forming SOX9^−^ *AKT-NICD* CCA nodules. However, we revealed that SOX9 is specifically required for the maintenance of fully developed *Akt-YAP1* but not *Akt-NICD* CCA tumors. This further supports our claim of overlapping and/or context/stage-dependent roles for SOX9 in liver cancer including CCA. 

Lastly, considering the significant therapeutic effect of *Sox9* ablation in advanced *Akt-YAP1* cHCC-CCA, similar to our *Sox9* CKO (acute deletion) data but contrasting with *Sox9* LKO (developmental deletion) in a preventive manner, there is a need to revisit tumor studies conducted using developmental *Cre* strains-mediated gene KO systems, which may overlook adaptation. This suggests the necessity for future studies to reevaluate therapeutic potentials, anticipating distinct response impact information, as preclinical data translates into relevance for human cancer patients.

## 5. Conclusions

Our study underscores the complex, context-dependent roles of SOX9 in the development and progression of cHCC-CCA. The contrasting outcomes observed between developmental *Sox9* elimination and tumor-specific acute *Sox9* ablation highlight the nuanced functions of SOX9 in liver cancer biology. Our findings advocate the need for careful consideration when using developmental *Cre* models in liver cancer research. Moreover, our results point to the potential for precise, personalized therapeutic strategies targeting SOX9 in treating clinical cHCC-CCA. Further research is essential to translate these preclinical insights into effective clinical treatments.

## Figures and Tables

**Figure 1 cells-13-01451-f001:**
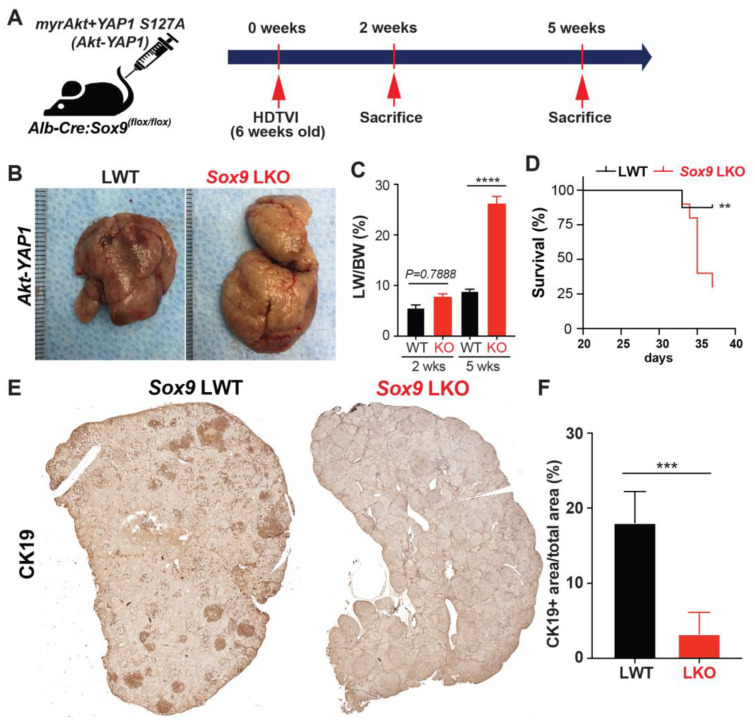
Chronic developmental deletion of *Sox9* switches the fate of *Akt-YAP1*-driven cHCC-CCA to aggressive HCC at the expense of CCA. (**A**) Experimental design illustrating plasmids used for HDTVI, mice used in study, and time-points analyzed. (**B**) Representative gross images from *Akt-YAP1*-injected *Sox9*-floxed mice (LWT) and *Akt-YAP1*-injected *Alb-Cre;Sox9^(f/f)^* liver-specific *Sox9* knockout or *Sox9* LKO mice showing tumor-laden enlarged livers in both cases. (**C**) LW/BW ratio depicts a significantly lower tumor burden in *Akt-YAP1 Sox9* LKO as compared to LWT at 5 weeks but not earlier than the 2-week time-point. (**D**) Kaplan–Meier survival curve showing a significant decrease in the survival of *Akt-YAP1* mice that were *Sox9* LKO as compared to LWT. (**E**) Representative tiled image of tumor-bearing livers at 5 weeks in *Akt-YAP1* LWT stained for CK19 IHC showing the CCA component of the positive cHCC-CCA staining. *Sox9 LKO* livers were full of circumscribed tumors that were negative for CK19 at the same time-point. (**F**) Quantification of CK19 IHC verifies significantly reduced staining in *Akt-YAP1 Sox9* LKO as compared to *Akt-YAP1* LWT at 5 weeks, as shown in E. Error bar: standard error of the mean; ** *p* < 0.01; *** *p* < 0.05; **** *p* < 0.0001.

**Figure 2 cells-13-01451-f002:**
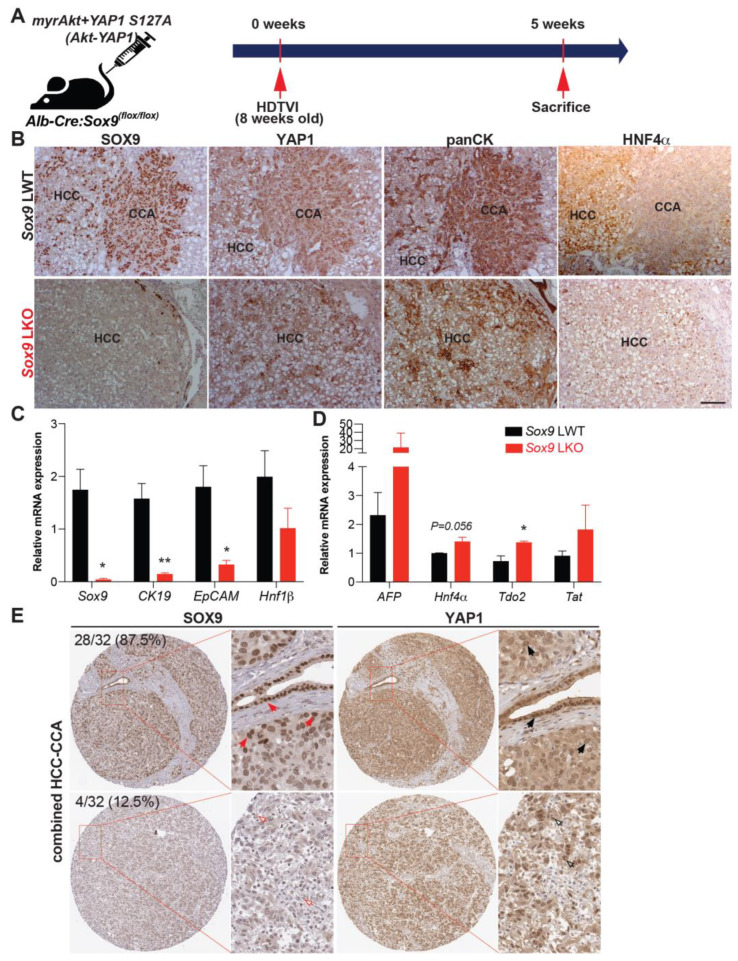
Absence of *Sox9* induces *Akt-YAP1*-mediated HC-derived panCK and HNF4α-positive HCC associated with liver progenitor cell characteristics. (**A**) Experimental design illustrating plasmids used for HDTVI, mice used in study, and time-points analyzed. (**B**) Representative serial sections of IHC images of 5-week *Akt-YAP1* LWT show CCA component to be positive for SOX9, YAP1, and panCK and HCC to be strongly positive for HNF4α, while in *Akt-YAP1 Sox9* LKO, no CCA was seen and HCC was negative for SOX9 but positive for HNF4α with panCK- and YAP1-positive cells interspersed in the tumor parenchyma. (**C**,**D**) qPCR data showing significantly decreased mRNA expression of *Sox9*, *CK19*, and *EpCAM* and significantly increased expression of *Tdo2* when comparing tumor-bearing livers in *Akt-YAP1* LWT and *Sox9* LKO models at 5 weeks. (**E**) Representative IHC staining for SOX9 and YAP1 depicting SOX9-low and nuclear YAP1-high or SOX9-high and YAP1-high cHCC-CCA from TMA (32 patients). TMA sections were enlarged for better view of nuclear expression of SOX9 and YAP1. Red arrows point to nuclear SOX9-high cells; black arrows point to nuclear YAP1-high cells; red empty arrows point to nuclear SOX9-negative cells; and black empty arrows point to nuclear YAP1-high cells. Percentage of patients positive for each combination are indicated. Scale bars: 100 µm; Error bar: standard error of the mean; * *p* < 0.05; ** *p* < 0.01.

**Figure 3 cells-13-01451-f003:**
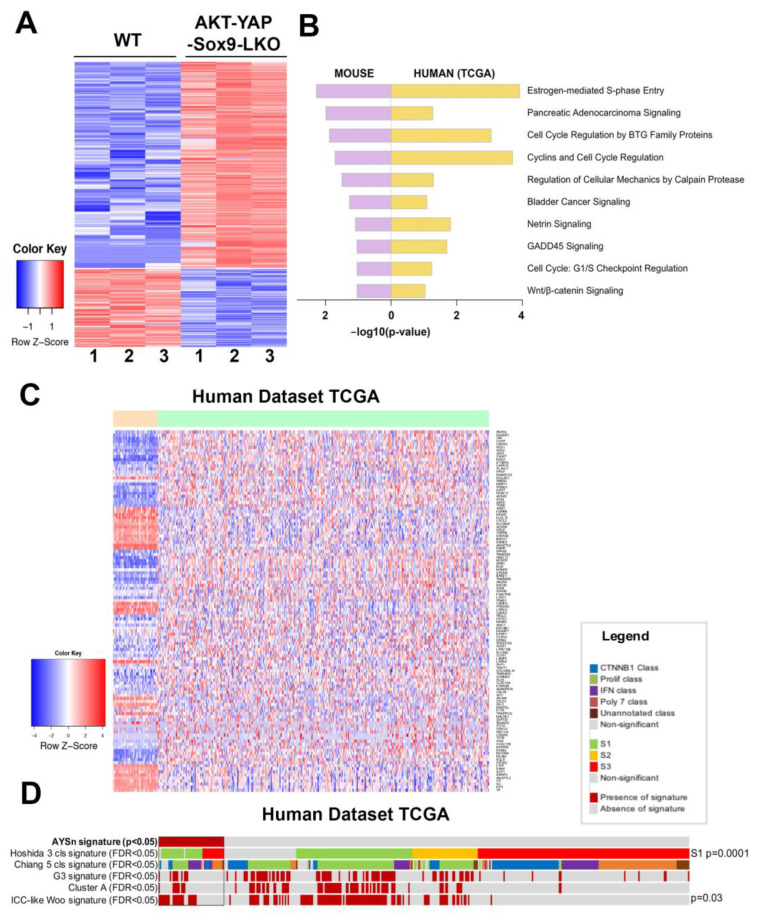
RNA-seq analysis of mouse models and comparison with human liver cancer studies. (**A**) Heatmap for the differentially expressed genes comparing LWT and the *Akt-YAP1 Sox9* LKO model. (**B**) Common top enriched pathways between mouse (*Akt-YAP1 Sox9* LKO) and human (TCGA, LIHC) study. (**C**) Heatmap of gene signatures in the TCGA LIHC that are selected by the mouse model (LWT vs. *Akt-YAP1 Sox9* LKO). (**D**) Nearest Template Prediction (NTP) analysis of the TCGA LIHC whole-tumor gene expression dataset using the *Akt-YAP1 Sox9* LKO signature (AYSn signature) captured 12% of HCC. This subgroup of patients is enriched in the S1 class [22] (*p* = 0.0015) and has an ICC-like signature [23] (*p* = 0.03).

**Figure 4 cells-13-01451-f004:**
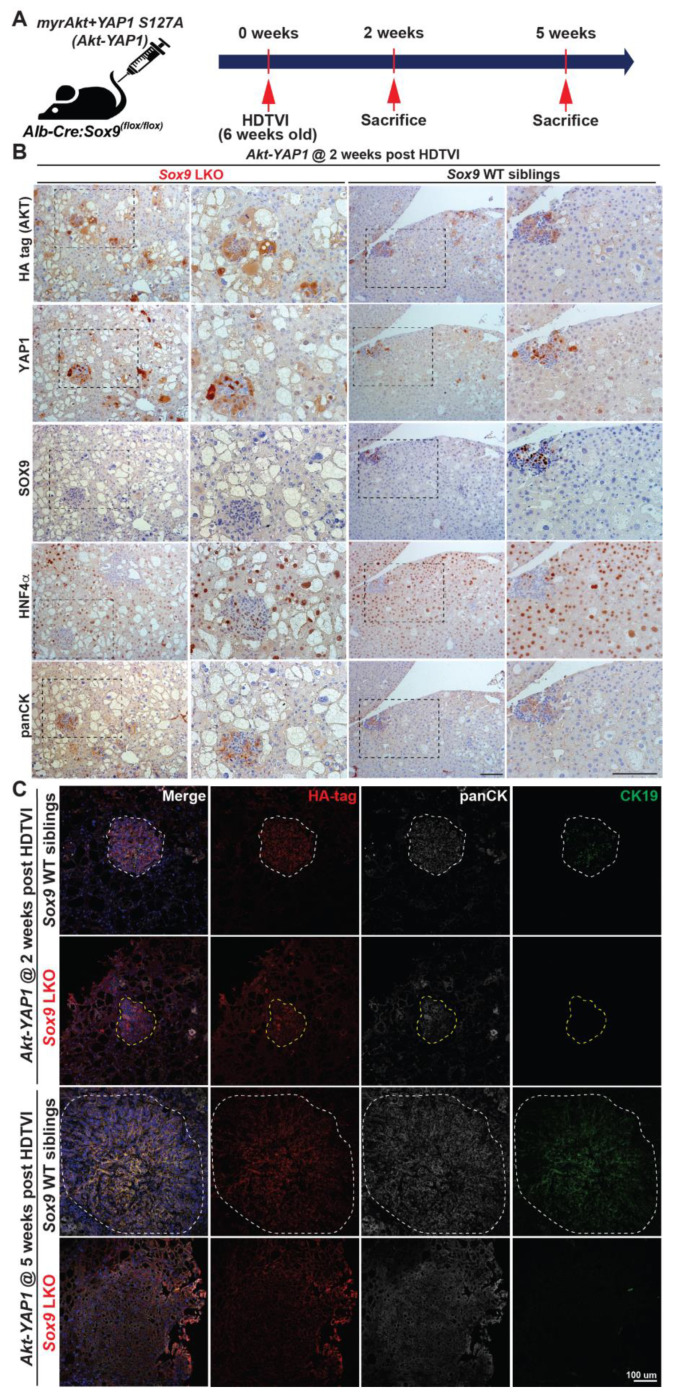
SOX9 is required for *Akt-YAP1*-mediated HC reprogramming into CK19^+^ mature CCA. (**A**) Experimental design illustrating plasmids used for HDTVI, mice used in study, and time-points analyzed. (**B**) Representative serial section IHC images of both *Akt-YAP1* LWT and *Akt-YAP1 Sox9* LKO show CCA-like components to be positive for SOX9, YAP1, panCK, and weak positive HNF4α with liver progenitor cell morphology (black dash lined) and HCC to be strongly positive for HNF4α at 2 weeks post-HDTVI. (**C**) Confocal images of immunofluorescence staining of LWT or *Sox9*-LKO livers at 2 and 5 weeks verify the essential roles for *Sox9* in CCA maturation. White dashed line points to panCK^+^;SOX9^+^;CK19^+^ CCA cells and yellow dashed line to panCK^+^;SOX9^−^;CK19^−^ LPC-like immature CCA cells. Scale bars: 100 µm.

**Figure 5 cells-13-01451-f005:**
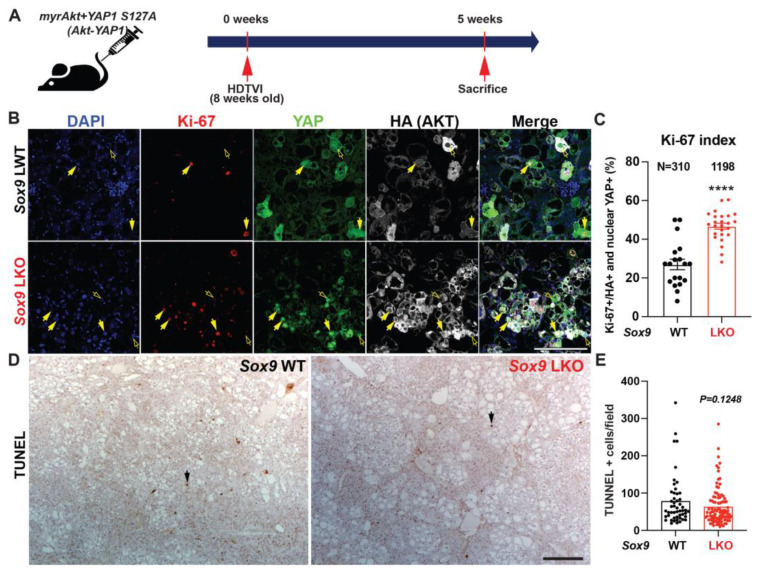
Developmental removal of *Sox9* promotes proliferation of *Akt-YAP1*-mediated liver cancer. (**A**) Experimental design illustrating plasmids used for HDTVI, mice used in study, and time-points analyzed. (**B**) Representative IF for Ki-67 (red), YAP1 (green), HA-tag (gray), and DAPI (blue) in liver sections from 5 weeks for *Akt-YAP1* LWT or *Sox9* LKO. Yellow arrows indicate Ki-67^+^;HA^−^tag+;YAP^+^ proliferating liver cancer cells and Yellow empty arrows point to Ki-67^−^;HA-tag^+^;YAP^+^ non-proliferating liver cancer cells. (**C**) The percentage of Ki-67-positive nuclei normalized to HA-tag-positive total tumor cell nuclei in representative images shown in B demonstrate significant increase in proliferation of transduced tumor cells in *Sox9* LKO as compared to LWT. (**D**) IHC for TUNEL to detect non-viable tumor cells shows comparable cell death was evident between *Sox9* LKO and LWT in *Akt-YAP1* model at 5-week time-point. Black arrows indicate nuclear TUNEL-positive apoptotic cells. (**E**) The number of TUNEL-positive nuclei normalized to field in representative images shown in D demonstrates comparable cell death in *Sox9* LKO as compared to WT. Scale bars: 100 µm; error bar: standard error of the mean; **** *p* < 0.0001.

**Figure 6 cells-13-01451-f006:**
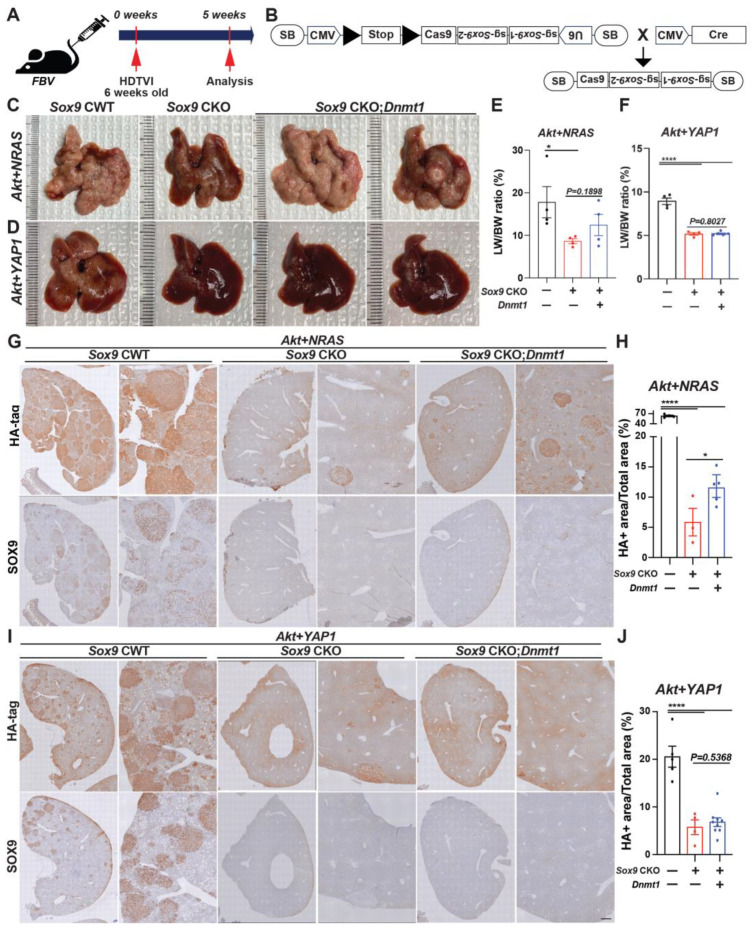
Acute *Sox9* deletion prevents the formation of combined HCC-CCA mediated by *Akt-YAP1* or *Akt-NRAS*, although *Akt-NRAS* tumor exhibit partial dependence on *Dnmt1*. (**A**) Experimental design illustrating plasmids used for HDTVI, mice used in study, and time-points analyzed. (**B**) A model illustrating the experimental design utilizing sleeping beauty transposon/transposase-CRISPR/Cas9-based inducible *Sox9* knockout plasmid. Representative gross images from *Akt-NRAS* (**C**) or *Akt-YAP1* (**D**)-injected WT (CWT), acute *Sox9*-knockout (CKO), and *Dnmt1*-injected Sox9-CKO (*Sox9* CKO-*Dnmt1*) livers showing tumor burden. (**E**) LW/BW ratio depicts significantly lower tumor burden in *Akt-NRAS* Sox9 CKO and *Akt-NRAS Sox9* CKO-*Dnmt1* mice as compared to CWT at 5 weeks. (**F**) LW/BW ratio depicts significantly lower tumor burden in *Akt-YAP1 Sox9* CKO and *Akt-YAP1 Sox9* CKO-*Dnmt1* mice as compared to CWT at 5 weeks. (**G**,**H**) Representative IHC images of tumor-bearing livers at 5 weeks in *Akt-NRAS* CWT, *Akt-NRAS Sox9* CKO, and *Akt-NRAS Sox9* CKO-*Dnmt1* liver stained for HA-tag and SOX9 showing cHCC-CCA component. HA-tag^+^ *Akt-NRAS* cHCC/CCA tumor burden was robustly abrogated in *Sox9* CKO livers while being slightly but significantly restored in *Dnmt1*-injected *Sox9*-CKO livers. (**I**,**J**) Representative IHC images of tumor-bearing livers at 5 weeks in *Akt-YAP1* CWT, *Akt-YAP1 Sox9* CKO, and *Akt-YAP1 Sox9* CKO-*Dnmt1* liver stained for HA-tag and SOX9 showing cHCC-CCA component. HA-tag^+^ *Akt-YAP1* cHCC/CCA tumor burden was robustly abrogated in both *Sox9* CKO and *Sox9* CKO-*Dnmt1* livers. Each dot in the graphs represent an individual mouse. Scale bars: 100 µm. Error bar: standard error of the mean; * *p* < 0.05; **** *p* < 0.0001.

**Figure 7 cells-13-01451-f007:**
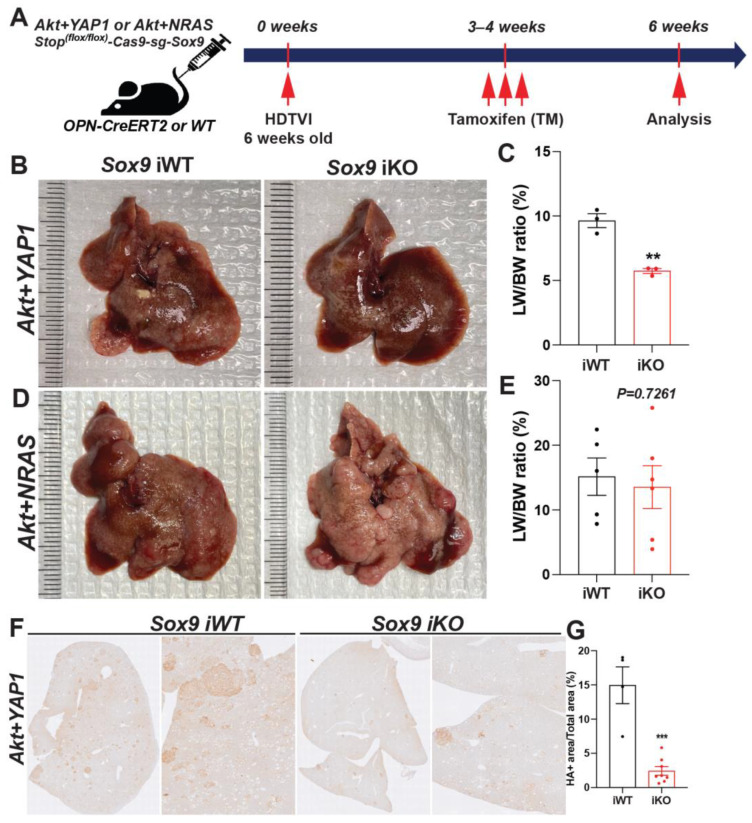
Therapeutic *Sox9* elimination reduces *Akt-YAP1-dependent* advanced combined HCC-CCA. (**A**) Experimental design illustrating plasmids used for HDTVI, Tamoxifen treatment, mouse strain used in study, and time-points analyzed. (**B**,**D**) Representative gross images from *Akt-YAP1* (**B**) or *Akt-NRAS* (**D**)-injected *OPN-CreERT2* mice along with *Stop^(f/f)−^Cas9-sg-empty* (*Sox9*-iWT) and *Stop^(f/f)^-Cas9-sg-Sox9* (*Sox9*-iKO) treated with Tamoxifen (100 mg/kg triple) displaying gross tumor burden. (**C**,**E**) LW/BW ratio depicts significantly lower tumor burden in *Akt-YAP1* (**C**) but not in *Akt-NRAS* (E) *Sox9* iKO as compared to iWT at 6 weeks. (**F**) Representative IHC image of tumor-bearing livers at 6 weeks in *Akt-YAP1* iWT stained for HA-tag IHC showing component of the cHCC-CCA staining. *Sox9* iKO livers significantly reduced HA-tag^+^ tumor burden. (**G**) Quantification of HA-tag IHC verifies significantly reduced staining in *Akt-YAP1 Sox9* iKO as compared to *Akt-YAP1 Sox9* iWT at 6 weeks, as shown in (**B**). Each dot in the graphs represent an individual mouse. Error bar: standard error of the mean; ** *p* < 0.01; *** *p* < 0.05.

## Data Availability

RNA-seq data have been submitted to the online database gene expression omnibus (GEO) accession ID: GSE200472.

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
