# Peer review of "Context-Dependent Distinct Roles of SOX9 in Combined Hepatocellular Carcinoma–Cholangiocarcinoma"

_cells, 2024, doi:10.3390/cells13171451_

Round 1
Reviewer 1 Report
Comments and Suggestions for Authors
This manuscript demonstrates the role of SOX9 in the development of a hepatocellular carcinoma and cholangiocarcinoma model. While the individual data are compelling, most of the experiments were performed in mouse models, which make the overall results difficult to understand. Experiments using various conditional knockout vectors are commendable, but I would like to see a summarized figure at the end that would show the results of the authors' experiments. The author should address to the following points.
In particular, biochemical experiments that can show hypotheses at the cellular level should be included, not only animal experiments (in addition to clinical data) in accordance with journal policy.
The authors should explain what DNMT1 is in this paper many times.
Author Response
Response to reviewer:
Comments and Suggestions for Authors
This manuscript demonstrates the role of SOX9 in the development of a hepatocellular carcinoma and cholangiocarcinoma model. While the individual data are compelling, most of the experiments were performed in mouse models, which make the overall results difficult to understand. Experiments using various conditional knockout vectors are commendable, but I would like to see a summarized figure at the end that would show the results of the authors' experiments. The author should address to the following points.
- A) Thank you, we have provided graphical abstract illustrates overall results and key message of this manuscript.
In particular, biochemical experiments that can show hypotheses at the cellular level should be included, not only animal experiments (in addition to clinical data) in accordance with journal policy.
- A) Would you please specify the additional experiments recommended to be included in this manuscript?
The authors should explain what DNMT1 is in this paper many times.
- A) Thank you for pointing that out. We have updated the manuscript by abbreviating and explaining the rationale for testing this gene in our context. Additionally, we have added sentences to further clarify the reasoning behind testing this gene.
Reviewer 2 Report
Comments and Suggestions for Authors
In this study, the authors have investigated the role of pathways involving the biliary-specific transcription factor SOX9 in liver cancer development, more precisely in combined hepatocellular carcinoma (HCC)/cholangiocarcinoma (CCA) tumors. The experimental design is reasonable, and the results support the conclusion.
Major comments
1. The title does not fit the manuscript content. A considerable effort has been made to elucidate the molecular bases of HCC/CCA carcinogenesis. It is correct to mention the therapeutic potential of the results in the study's conclusion, but it is far from justified to include it in the title, which is quite misleading in its present form.
2. The text of this manuscript is very wordy and not fluently presented. There is considerable redundancy, particularly in the Discussion section. An effort to simplify and more clearly present the findings is strongly recommended.
Minor suggestions
1. The authors must revise the correct use of nomenclature for genes (italic font) and proteins (plain font), which is sometimes incorrect (e.g., lines 62, 123, 391, and so on).
2. Please use italic font for in vitro and in vitro (lines 371-372).
3. Once defined, always use the abbreviation (e.g., line 98).
4. Place space between values and units (e.g., line 340, 100mg/kg)
Comments on the Quality of English LanguageSome corrections, already listed in my report, are required.
Author Response
Comments and Suggestions for Authors
In this study, the authors have investigated the role of pathways involving the biliary-specific transcription factor SOX9 in liver cancer development, more precisely in combined hepatocellular carcinoma (HCC)/cholangiocarcinoma (CCA) tumors. The experimental design is reasonable, and the results support the conclusion.
- A) Thank you for the comments.
Major comments
- The title does not fit the manuscript content. A considerable effort has been made to elucidate the molecular bases of HCC/CCA carcinogenesis. It is correct to mention the therapeutic potential of the results in the study's conclusion, but it is far from justified to include it in the title, which is quite misleading in its present form.
- A) Thank you for the suggestion; we agree with your point and have revised our title to “Context-Dependent Distinct Roles of SOX9 in Combined Hepatocellular Carcinoma-Cholangiocarcinoma.”
- The text of this manuscript is very wordy and not fluently presented. There is considerable redundancy, particularly in the Discussion section. An effort to simplify and more clearly present the findings is strongly recommended.
- A) Thank you for the suggestion. We have revised and polished the English throughout the entire manuscript to improve clarity and readability. All changes have been highlighted in red in the revised version.
Minor suggestions
- The authors must revise the correct use of nomenclature for genes (italic font) and proteins (plain font), which is sometimes incorrect (e.g., lines 62, 123, 391, and so on).
- Please use italic font for in vitro and in vitro (lines 371-372).
- Once defined, always use the abbreviation (e.g., line 98).
- Place space between values and units (e.g., line 340, 100mg/kg)
A) Thank you for pointing that out. We've revised the entire manuscript to correct the font, abbreviations, and other details. All changes have been highlighted in red in the revised version.
Reviewer 3 Report
Comments and Suggestions for Authors
In this manuscript, Park et al show that developmental Sox9 and tumor-specific deletion of Sox9 in livers with Akt-YAP1 and Akt-NRAS driven cHCC-CCA tumors resulted in reduction of tumor burden. The experiment mimicking therapeutic Sox9 elimination using the OPN-CreERT2 strain to acutely delete Sox9 in existing tumors is probably the most interesting piece of data, and showed tumor reduction in both tumor models. Thile the experiments were elegantly and comprehensively done, the contribution of this manuscript is only incremental because Liu et al. (DOI:https://doi.org/10.1016/j.jhep.2021.11.010) already showed the interplay between Yap and Sox9 and the potential of Sox9 inhibition for targeted therapy. Nonetheless, this piece of work will nicely complement previous works and will cement the idea that SOX9 elimination may hold promise as a therapeutic approach for cHCC-CCA. Thus, this manuscript should be accepted provided that the authors respond to the critique below.
1. Alb-Cre recombines floxed alles during early hepatoblast development, which results in mutant adult hepatocytes and cholangiocytes, which will result in livers with bile duct defects. How does the lack of Sox9 affect biliary duct development, cholestasis and tumor formation?
2. The authors say: “Importantly, CCA-like 222 nodules with Akt-YAP1 transduction (HA-tag+; nuclear YAP1+) and the retention of inter- mediate LPC morphology, along with co-expression of HC marker HNF4α and BEC 224 marker panCK, were successfully developed in both LWT and Sox9-LKO livers whereas the presence or absence of SOX9 in these nodules differed between the respective livers (Fig.4 black dash lined). This data suggests that while SOX9 is not necessary for YAP1-227 driven biliary reprogramming, it is essential for the survival and maintenance of these 228 CCA-like tumor nodules.”
How do they conclude this? The authors say that Sox9 is NOT necessary for BEC reprograming, yet in figure 4 Sox9 KO tumors do not express BEC markers. panCK while expressed in BECs, it is also expressed in undifferentiated progenitor-like cells, thus this data does not support the fact that Sox9 is dispensable for BEC reprogramming in tumors driven by AKT-YAP. A double immunofluorescent staining of HNF4a and panCK as double positive cells will show progenitor phenotype rather than BEC reprogramming. For BEC reprogramming, the authors need to show that HA positive cells are negative for HNF4a and positive for any BEC marker (OPN, CK19, EPCAM, Mucin, etc). Also, the text requires to be revised to simplify its understanding.
3. How does the developmental deletion of Sox9 allow tumor formation while the adult tumor specific deletion of Sox9 prevents tumor formation (AKT-YAP model)? Is this because Sox9 deletion in BECs (AlbCre) cause cholestasis and this potentiates HCC development from oncogene expressing cells? Is tumor formation restored in the model that uses CrisprCas9 if the livers are cholestatic by bile duct ligation? What happens when rather than deleting Sox9 by Alb-cre during development, Sox9 is specifically deleted in adult hepatocytes with AAV9-Cre?
4. If not cholestasis, is this effect due to cell competition and a novel function of Sox9 in the fitness of normal and cancer liver cells? That is, if Sox9 is solely deleted from cancer cells, peritumoral hepatocytes outcompete Sox9 cancer cells, but this effect is abrogated when hepatocytes are also mutant.
5. Why Dnmt1 re-expression in Sox9 CKO liver restored tumor formation in Akt-NRAS but not Akt-YAP? Is this difference due to the pathways or due to the type of tumor?
6. The inducible deletion of Sox9 in grown tumors showed a sticking reduction in tumor load, which suggest that Sox9 could be consider as a potential target for anti-cancer therapy. However developmental deletion of Sox9 causes Biliary Atresia while deletion of Sox9 in adult BECs causes Cholangiocyte dysfunction, fibrosis and liver injury. Thus, how can Sox9 be considered as a therapeutic target if the side effects are so severe? Discuss how Sox9 could be targeted for cancer therapy.
Comments on the Quality of English LanguageMinor revisions should be done.
Author Response
Comments and Suggestions for Authors
In this manuscript, Park et al show that developmental Sox9 and tumor-specific deletion of Sox9 in livers with Akt-YAP1 and Akt-NRAS driven cHCC-CCA tumors resulted in reduction of tumor burden. The experiment mimicking therapeutic Sox9 elimination using the OPN-CreERT2 strain to acutely delete Sox9 in existing tumors is probably the most interesting piece of data, and showed tumor reduction in both tumor models. Thile the experiments were elegantly and comprehensively done, the contribution of this manuscript is only incremental because Liu et al. (DOI:https://doi.org/10.1016/j.jhep.2021.11.010) already showed the interplay between Yap and Sox9 and the potential of Sox9 inhibition for targeted therapy. Nonetheless, this piece of work will nicely complement previous works and will cement the idea that SOX9 elimination may hold promise as a therapeutic approach for cHCC-CCA. Thus, this manuscript should be accepted provided that the authors respond to the critique below.
A) Thank you for the favorable comments and expertise perspectives.
- Alb-Cre recombines floxed allele during early hepatoblast development, which results in mutant adult hepatocytes and cholangiocytes, which will result in livers with bile duct defects. How does the lack of Sox9 affect biliary duct development, cholestasis and tumor formation?
A) We thank the reviewer for this comment. Based on our knowledge and a thorough literature search, Alb-Cre;Sox9 KO (Sox9 LKO) induces a delay in bile duct development, which is compensated by Sox4 (PMID: 19403103; 26013091; 33512716). Over the long term, only a mild cholestasis phenotype has been observed at 12 months of age, evidenced by a slight elevation in ALP but no significant difference in total bilirubin levels (PMID: 33512716). Given that 12 months in mice is roughly equivalent to 40 years in humans, this scenario does not align with our study design. Since our study timeline spans 6-12 weeks, we do not expect a severe bile duct defect due to Sox9 deletion in cholangiocytes to influence liver cancer formation.
In other projects within our lab, we have simultaneously ablated Sox9 and other key biliary-specific transcription factors from adult cholangiocytes without observing any cholestasis symptoms or differences in serum values, including ALT, ALP, AST, or total bilirubin levels, compared to control animals. However, we have observed lethal cholestasis when eliminating YAP and TAZ in cholangiocytes, as previously reported (PMID: 33127392). Thus, we believe that cholestasis is not a significant factor affecting liver cancer formation following Sox9 deletion. Additionally, we have not observed any cholestasis symptoms in baseline Sox9 LKO animals prior to HDTVI-induced liver cancer.
- The authors say: “Importantly, CCA-like 222 nodules with Akt-YAP1 transduction (HA-tag+; nuclear YAP1+) and the retention of inter- mediate LPC morphology, along with co-expression of HC marker HNF4α and BEC 224 marker panCK, were successfully developed in both LWT and Sox9-LKO livers whereas the presence or absence of SOX9 in these nodules differed between the respective livers (Fig.4 black dash lined). This data suggests that while SOX9 is not necessary for YAP1-227 driven biliary reprogramming, it is essential for the survival and maintenance of these 228 CCA-like tumor nodules.”
How do they conclude this? The authors say that Sox9 is NOT necessary for BEC reprograming, yet in figure 4 Sox9 KO tumors do not express BEC markers. panCK while expressed in BECs, it is also expressed in undifferentiated progenitor-like cells, thus this data does not support the fact that Sox9 is dispensable for BEC reprogramming in tumors driven by AKT-YAP. A double immunofluorescent staining of HNF4a and panCK as double positive cells will show progenitor phenotype rather than BEC reprogramming. For BEC reprogramming, the authors need to show that HA positive cells are negative for HNF4a and positive for any BEC marker (OPN, CK19, EPCAM, Mucin, etc). Also, the text requires to be revised to simplify its understanding.
A) We appreciate the reviewer’s expert opinion. As suggested, we performed immunofluorescence for CK19 on both LKO and LWT mice at 2 and 5 weeks post-HDTVI. As described in the manuscript, panCK+ LPC-like immature nodules were observed in both LWT and LKO livers at 2 weeks. However, as the reviewer pointed out, we found that a subset of these nodules was CK19+ mature CCA nodules, which were present only in LWT livers at this early phase (2 weeks post-HDTVI). This suggests that Sox9 is required for the further transformation and maturation of these nodules into CK19+ mature CCA cells. By 5 weeks, these nodules progressed to mature CCA only in Sox9 WT livers and not in KO livers; in Sox9 LKO livers, PanCK+; HA+ immature CCA nodules had completely disappeared by 5 weeks post-HDTVI.
Based on our observations, we conclude that Sox9 is essential for the maturation and maintenance of these immature CCA tumor cells in the AY model but is not necessary for the formation of HC-derived LPC-like immature CCA precursor nodules. Accordingly, we have included co-IF data into Figure 4C and revised our manuscript to reflect these findings more accurately. Thank you again for this valuable insight.
- How does the developmental deletion of Sox9 allow tumor formation while the adult tumor specific deletion of Sox9 prevents tumor formation (AKT-YAP model)? Is this because Sox9 deletion in BECs (AlbCre) cause cholestasis and this potentiates HCC development from oncogene expressing cells? Is tumor formation restored in the model that uses CrisprCas9 if the livers are cholestatic by bile duct ligation? What happens when rather than deleting Sox9 by Alb-cre during development, Sox9 is specifically deleted in adult hepatocytes with AAV9-Cre?
A) We thank the reviewer for these comments and questions. We were also surprised by these results and find it challenging to provide a clear explanation. As briefly mentioned above, we believe that unless aging is factored in, the effect of Sox9 singular deletion in cholangiocytes appears minimal. Specifically, since our study period is less than 3 months, we do not think that the absence of Sox9-mediated cholestasis is the primary cause of liver cancer promotion. Instead, we believe that the role of Sox9 in HC-to-CCA/HCC transformation may be dependent on the stage of the disease and/or the specific oncogenic driving factors. Additionally, the discrepancy between chronic developmental deletion and acute deletion may be due to the developmental adaptation in response to the absence of Sox9, potentially compensated by factors like Sox4 in bile duct formation.
Indeed, we have confirmed that both tumor-specific acute deletion using a Cre expression plasmid and AAV8-Cre-mediated HC-specific ablation of Sox9 delay Akt-NICD CCA formation, while showing no therapeutic effect on advanced Akt-NICD CCA. This suggests that Sox9’s role in liver cancer may be stage-dependent and influenced by different oncogenic drivers.
We would be eager to perform AAV8;Sox9 KO experiments in our AY and ANRAS cHCC-CCA models, as recommended. However, since the Sox9(f/f) strain has already been bred with other floxed alleles and the original allele is cryopreserved at the Jax facility, it would not be practical to conduct AAV8-Cre;Sox9(f/f) experiments for this manuscript. Nevertheless, we previously reported that acute Sox9 deletion in the Akt-NICD CCA model simply delays CCA formation, resulting in delayed Sox9-;CK19+ CCA formation, indicating a dispensable role for Sox9 in HC-to-CCA reprogramming in the Akt-NICD context.
- If not cholestasis, is this effect due to cell competition and a novel function of Sox9 in the fitness of normal and cancer liver cells? That is, if Sox9 is solely deleted from cancer cells, peritumoral hepatocytes outcompete Sox9 cancer cells, but this effect is abrogated when hepatocytes are also mutant.
A) Although CCA tumor might compete with surrounding hepatocytes, given that Sox9 is a nuclear-specific transcription factor, we more believe that developmental adaptation is more likely the reason for the distinct response to Sox9 ablation. As mentioned briefly above and in the discussion section, we observed delayed Akt-NICD-CCA formation in both tumor-specific Sox9 ablation and in AAV8-Sox9 LKO animals.
- Why Dnmt1 re-expression in Sox9 CKO liver restored tumor formation in Akt-NRAS but not Akt-YAP? Is this difference due to the pathways or due to the type of tumor?
A) Thank you for this question. In our previous study (PMID: 35550144), we demonstrated that the YAP-DNMT1-HC-specific transcription factor signaling cascade permits Akt-NICD-mediated HC-to-CCA reprogramming. We also showed that DNMT1 inhibition eliminates the Akt-YAP1-CCA region while leaving the HCC region intact, indicating that DNMT1 is required for AKT-YAP-mediated HC-to-CCA reprogramming but is dispensable for HC transformation into HCC. Since the effect of Sox9 deletion in the AY model is similar to that of DNMT1 inhibition, we investigated whether there is any crosstalk between Sox9 and DNMT1 in the AY/ANRAS models. Given that DNMT1 is a direct downstream target of YAP, we observed that re-expression does not restore CCA, suggesting that Sox9 is not associated with DNMT1 in the Akt-YAP cHCC-CCA model. However, in the Akt-NRAS model, DNMT1 is independently regulated by Yap1 and partially crosstalk with Sox9, indicating distinct CCA-driving pathways between these two models.
The inducible deletion of Sox9 in grown tumors showed a sticking reduction in tumor load, which suggest that Sox9 could be consider as a potential target for anti-cancer therapy. However developmental deletion of Sox9 causes Biliary Atresia while deletion of Sox9 in adult BECs causes Cholangiocyte dysfunction, fibrosis and liver injury. Thus, how can Sox9 be considered as a therapeutic target if the side effects are so severe? Discuss how Sox9 could be targeted for cancer therapy.
A) Again, based on our knowledge and observations, Sox9 singular deletion in hepatocytes and/or cholangiocytes does not lead to noticeable or severe bile duct injury. In contrast, genetic co-deletion of YAP/TAZ results in lethal cholestasis, suggesting that Sox9 remains a relatively safe therapeutic target. Additionally, similar to how chemical YAP/TAZ inhibitors are less toxic than genetic elimination, a chemical approach to repressing Sox9 activity should be considered as a therapeutic option for liver cancer and PDAC, which exhibit similar molecular and pathological characteristics. It's also worth noting that YAP/TAZ inhibitors are currently in clinical trials for various tumors, indicating the potential for clinical applications of Sox9-targeting therapies as well.
Round 2
Reviewer 3 Report
Comments and Suggestions for Authors
The authors satisfactorily answered all my concerns. The paper can be accepted without any further changes.